# Hyperfine-phonon spin relaxation in a single-electron GaAs quantum dot

Leon C. Camenzind[1], Liuqi Yu[1], Peter Stano[2,3,4], Jeramy D. Zimmerman [5,6], Arthur C. Gossard[5], Daniel Loss[1,2] & Dominik M. Zumbühl[1]

Understanding and control of the spin relaxation time $T_1$ is among the key challenges for spin-based qubits. A larger $T_1$ is generally favored, setting the fundamental upper limit to the qubit coherence and spin readout fidelity. In GaAs quantum dots at low temperatures and high in-plane magnetic fields **B**, the spin relaxation relies on phonon emission and spin–orbit coupling. The characteristic dependence $T_1 \propto B^{-5}$ and pronounced B-field anisotropy were already confirmed experimentally. However, it has also been predicted 15 years ago that at low enough fields, the spin–orbit interaction is replaced by the coupling to the nuclear spins, where the relaxation becomes isotropic, and the scaling changes to $T_1 \propto B^{-3}$. Here, we establish these predictions experimentally, by measuring $T_1$ over an unprecedented range of magnetic fields—made possible by lower temperature—and report a maximum $T_1 = 57 \pm 15$ s at the lowest fields, setting a record electron spin lifetime in a nanostructure.

[1] Department of Physics, University of Basel, Klingelbergstrasse 82, 4056 Basel, Switzerland. [2] Center for Emergent Matter Science, RIKEN, Saitama 351-0198, Japan. [3] Department of Applied Physics, School of Engineering, University of Tokyo, 7-3-1 Hongo, Bunkyo-ku, Tokyo 113-8656, Japan. [4] Institute of Physics, Slovak Academy of Sciences, 845 11 Bratislava, Slovakia. [5] Materials Department, University of California, Santa Barbara, CA 93106, USA. [6] Present address: Physics Department, Colorado School of Mines, Golden, CO 80401, USA. These authors contributed equally: Leon C. Camenzind, Liuqi Yu. Correspondence and requests for materials should be addressed to D.M.Z. (email: dominik.zumbuhl@unibas.ch)

The decay of the energy stored in the qubit defines the relaxation time $T_1$. In qubits based on electronic spins, it corresponds to the relaxation of spin—a longstanding topic of research in semiconductors. The suppression of this process in a confined system compared to the bulk[1] makes quantum dot spin qubits a serious candidate for a quantum technology platform[2–4]. For spin qubits, the energy splitting is due to the Zeeman term of an applied magnetic field $B$. The requirement for a sizable splitting, necessary for many of the protocols to initialize, measure, or manipulate spin qubits[5–8], then imposes limitations on $T_1$, which in turn might influence these protocols in a profound way[9–11]. This further motivates investigations of mechanisms and fundamental limits of the spin relaxation in quantum dots.

To understand this process in a GaAs quantum dot spin qubit, one needs to consider that it involves the dissipation of both energy and angular momentum, i.e., spin. The former proceeds by emission of a phonon. Considering, for simplicity, long-wavelength three-dimensional bulk phonons, one gets the spin relaxation rate $W \equiv T_1^{-1} \propto B^3 d^2$ for piezoelectric and $W \propto B^5 d^2$ for deformation potential phonons, where $d$ is the dipole moment matrix element between the initial and final state of the transition. For typical Zeeman energies, piezoelectric phonons dominate. Since the initial and final states are opposite in spin, a non-zero dipole element can only arise due to some spin-dependent interaction. In GaAs, the two most relevant ones are the spin–orbit and hyperfine (HF) interactions. Their essential difference here is the time-reversal symmetry of the spin–orbit interaction (SOI), which also implies $T_2 = 2T_1$[12]; there is no such relation for the HF effects. While the HF interaction induces a $B$-independent moment, the time reversal symmetry of the SOI results, through the Van-Vleck cancellation, in an additional magnetic field proportionality, $d^2 \propto B^2$. Putting these pieces together, the SOI, with $W \propto B^5$, will dominate at high fields, and HF, with $W \propto B^3$, at low fields. For the parameters of typical surface gate defined GaAs dots, the crossover is predicted at around 1–2 T. We estimate that in natural silicon the crossover would happen at magnetic fields roughly hundred times smaller.

Beyond field scaling, the SOI with competing Rashba and Dresselhaus terms results in a strong dependence of spin relaxation on the direction of the applied magnetic field in the plane of the two-dimensional (2D) gas—the spin relaxation anisotropy[12–14]. The HF mechanism, on the other hand, is isotropic[15], even for a dot shape which breaks circular symmetry. These two hallmark features together—isotropic behavior and $B^3$ scaling—constitute a unique fingerprint of the HF relaxation mechanism. Note that the phonon-assisted inelastic transition is fundamentally different from the elastic electron-nuclear spin flip-flop, which is strongly suppressed due to the pronounced mismatch of the electron and nuclear Zeeman energy for fields above a few mT[16].

Even though the HF-assisted mechanism of spin relaxation was predicted early on[15], experimental observation has remained elusive so far for a number of reasons: rather low fields below 1 T are required to reach the HF limit. For a spin doublet, only energy selective spin-readout is available, thus requiring rather low electron temperatures below 100 mK to keep the Zeeman splitting well above the thermal broadening. To check for the direction dependence of relaxation, suitable piezo rotator control over the applied field direction is required, but this has only relatively recently become available. Finally, very long $T_1$-times far exceeding 1 s are predicted at such low fields, posing a formidable challenge on the long-term stability and control of a semiconductor nanostructure. Here, we overcome these difficulties by employing a very stable 2D gas and implementing active feedback procedures to keep the energy levels aligned with sub-microvolt precision over days (Supplementary Note 2). Specially developed Ag-epoxy filters[17] provide an electron temperature of ~60 mK—more than a factor of two lower than before[18]. Using these advances, we show isotropic relaxation combined with a $T_1 \propto B^{-3}$ scaling at low magnetic fields, thus demonstrating the hallmark signatures of hyperfine-phonon spin relaxation. At the lowest fields, we find $T_1 = 57 \pm 15$ s—a new record spin lifetime in a nanostructure. The error range specified here and elsewhere in this work is one standard deviation, as obtained from fitting.

## Results

**Quantum dot orbitals.** We use a flexible gate layout (Fig. 1a) to shape a nearly circularly symmetric dot and set up a cryogenic piezo-rotator to apply almost perfectly aligned in-plane fields (Supplementary Note 1) up to 14 T with arbitrary angle $\phi$ with respect to the [100] crystal direction (Fig. 1b). The rotator capability allows us to probe the dot orbitals and their shape in large magnetic fields using the established technique of pulsed-gate orbital excited state spectroscopy[18]. Figure 1c displays two excited states, shown in green and blue, for field applied along the $\hat{x}$ direction. While one state clearly moves down in energy (blue) with increasing field, the other one remains unaffected (green). Since only electron motion perpendicular to the applied field is affected by it, the $B$-invariant energy thus corresponds to the excitation along the $\hat{x}$ direction, justifying labels as shown in Fig. 1c[19,20]. When the sample is rotated by 90°, the excitations' roles swap and the blue line becomes invariant (Fig. 1d). Such striking behavior, including further $B$-directions, is reproduced by an anisotropic harmonic oscillator model[21,22], which confirms that the quantum dot main axes are well aligned with the $\hat{x}$ and $\hat{y}$ directions. This essential information about the dot orbitals makes possible a detailed understanding of all measurements, reproducing the measured $T_1$ quantitatively by numerics using a single set of parameters without phenomenological constants (see Methods for details).

**Spin-orbit induced spin relaxation anisotropy.** With a full orbital model at hand, we now turn to spin relaxation measurements, done by cycling the dot through ionization, charge and relax, and read-out configuration, as depicted in Fig. 2a. Averaging over many thousand cycles, we obtain the spin excited state probability $P_e$ as a function of the waiting time $t_w$, the time the electron was given to relax into the spin ground state. A few examples are plotted over four orders of magnitude in $t_w$ in Fig. 2b at a magnetic field of 4 T. All such curves fit very well to the sum of two exponentials, from which we reliably extract the spin relaxation rate $W \equiv T_1^{-1}$ (see Supplementary Note 3 for more details). A pronounced dependence of $W$ on the direction of the magnetic field is observed, as displayed in Fig. 3a as a function of the field angle $\phi$. A modulation of $W$ by a factor of ~16 is found, with minimal relaxation rate along the $\hat{y}$ direction.

This pronounced anisotropy is rooted in a combination of the dot shape asymmetry and the interference of the Rashba and Dresselhaus SOI terms. The latter can qualitatively be understood from the dependence of the total effective spin–orbit magnetic field on the direction of the electron momentum (Supplementary Note 5). First derived for symmetric quantum dots[12], the spin relaxation anisotropy due to the dot shape asymmetry was also soon included in a theoretical generalization[13]. The shape-induced contribution to the anisotropy of $W$ is well known here from the orbital spectroscopy and found to be small. Thus, the anisotropy here is largely due to the SOI, and given the precisely measured orbital energies, it is possible to extract the SOI coupling strengths by fitting the model (see Methods for details). The best fit delivers a ratio $\alpha/\beta \sim 1.6$ and a spin–orbit length $l_{so} \approx 2.1$ μm setting the overall strength of the SOI. These values are well in-line with previous reports for GaAs structures[18,23,24]. We note that $\alpha$ and $\beta$ are found to have the same sign for the 2D

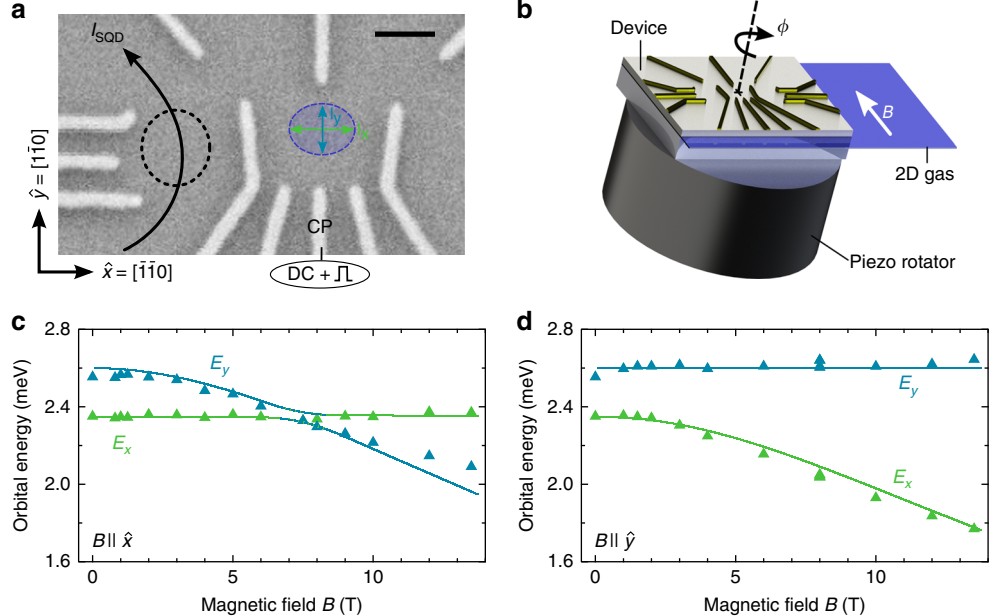

**Fig. 1** Quantum dot setup and orbital spectroscopy. **a** Scanning electron microscope image of a co-fabricated lateral, surface gate defined quantum dot. The single electron wave function is indicated by the blue ellipse (not to scale) and is tunnel coupled to the left reservoir only (no tunneling to right lead). An adjacent dot (black circle) serves as a real-time charge sensor, operated in Coulomb blockade for better sensitivity. Sub-microsecond pulses are applied on the center plunger CP. The scale bar corresponds to 200 nm. **b** Measurement setup with sample on a piezo-electric rotator allowing change of the direction of the in-plane magnetic field (up to 14 T) with respect to the crystal axis [100], specified by the angle $\phi$. **c**, **d** Energies of the two lowest orbital excited states, $E_x$ and $E_y$, measured with respect to the ground state, as a function of the magnetic field applied along $\hat{x}$ direction (**c**) and $\hat{y}$ direction (**d**). Triangles are measured data, solid curves are numerics (see text)

material used. Without knowledge of the orbital energies, the SOI parameters cannot be directly determined from $T_1$[14,25,26].

**Hyperfine-phonon spin relaxation.** A very long $T_1$ time can be achieved by reducing the magnetic field strength and orienting the magnetic field along the crystalline axis with minimal SOI field. Therefore, we carried out the same anisotropy measurements at 1.25 T. Indeed, $T_1$ times longer than 1 s are obtained. Interestingly, in contrast to the measurements at 4 T, around the $\hat{y}$ direction with minimal $W$, the measured spin relaxation rate $W$ (black markers) is seen to be almost a factor of three larger than the calculated SOI rate (red curve, Fig. 3b). This is far beyond the error bars, and indicates an additional spin relaxation channel beyond SOI-mediated phonon emission.

Because the dot orbitals are characterized, the HF contribution can be quantified by numerics (Methods). As shown in Fig. 3a, at $B = 4$ T the microscopic model predicts that the HF contribution (orange curve) is 1 to 2 orders of magnitude smaller than the one due to the SOI (red curve), and is therefore not observable experimentally. In comparison, at $B = 1.25$ T, as shown in Fig. 3b, the SOI model alone is unable to explain the data, but fits very well when the nuclei are included (purple curve), particularly now capturing the minimum close to the $\hat{y}$-direction very well. Backed by numerics, we thus conclude that this seemingly subtle feature in the angular modulation of $W$ actually constitutes the first evidence of the HF relaxation mechanism.

To substantiate this claim, we measure the field magnitude dependence of $W$. In Fig. 4a we compare two sets, for the magnetic field along the $\hat{x}$ and $\hat{y}$ direction, where the effects of the nuclei with respect to SOI are, respectively, maximal and minimal. The two curves indeed show pronounced differences. With the field along the $\hat{x}$-direction, the relaxation follows the $B^5$ scaling quite well over the entire range of the measured magnetic fields. Thus, for the $\hat{x}$ direction, the relaxation is dominated by the

SOI for the full field range. In contrast, for fields along $\hat{y}$, there is a crossover around 2 T with a change of the power law scaling from roughly $B^5$ at high fields to $B^3$ at low fields, corresponding to a crossover from SOI to HF dominated relaxation.

Some comments are in place. First, dynamic nuclear spin polarization would distort the power laws. The absence of nuclear spin polarization in our measurements is guaranteed by the slowness of electron spin transitions at low fields and is an important advantage over experiments exploiting Pauli spin blockade in double dots. Second, the only remaining discrepancy of data and model is seen at high fields (see the blue data points and theory curve in Fig. 4a for $B \gtrsim 6$ T). This saturation is predicted in perturbative calculations[12,27,28] and exact numerics[13,29], including our model here, but it is not observed in our data. The explanation needs further investigations. Nevertheless, the issue is irrelevant for the nuclear-induced relaxation taking place at much smaller fields and longer times. Finally, we note a $T_1$ time of $57 \pm 15$ s for a magnetic field of 0.6–0.7 T along $\hat{y}$, where the range represents the error from fitting (Supplementary Note 3). To our knowledge, this is the longest $T_1$ time reported to date in a nanoelectronic device[10,18,26].

This all being said, we stress that the simple observation of a change in the power law scaling of $W \propto B^3$ is not sufficient as a proof of its HF origin. It could be that the phonons as an energy dissipation channel are replaced by another bath, e.g., charge noise or an ohmic bath also leads to a $B^3$ dependence[30–32]. The absence of deviations in the scaling of the $B||\hat{x}$ data indicates that phonons are responsible for the energy dissipation throughout and the crossover in the $\hat{y}$ data is not related to a specific value of $W$, or transition energy. Also, if the SOI remained as the mixing mechanism and the energy dissipation channel instead were to change, then the spin relaxation anisotropy, quantified by the ratio $W_X/W_Y$, would remain large at low fields. However, as

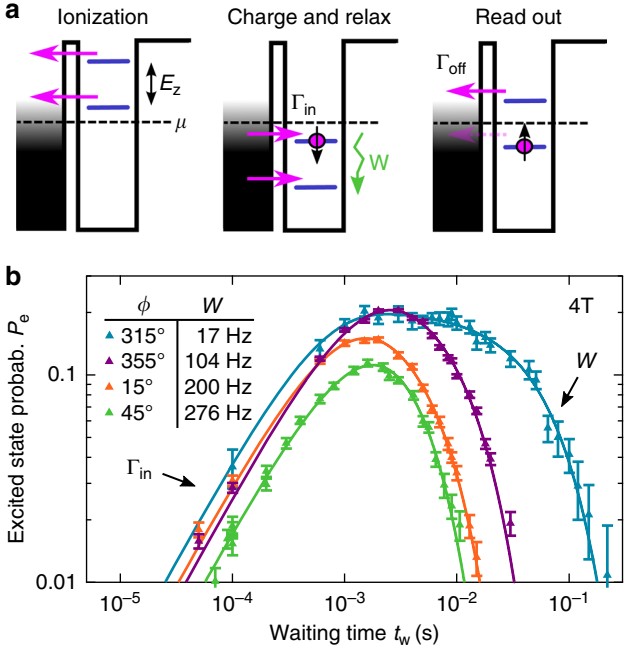

**Fig. 2** Spin relaxation measurement. **a** Three step pulse scheme, shifting dot levels with gate-voltage pulses: First, during "ionization", the dot is emptied. Second, in "charge and relax", an electron is loaded and if the spin is down, i.e., in the excited spin state, it relaxes with rate $W$ during the waiting time $t_w$. Third, spin-charge conversion is used in "read-out" to detect the spin state: the spin-down electron only will tunnel off the dot, which is detected by the charge sensor. The spin relaxation rate $W$ is extracted from the dependence of the probability $P_e$ to find the spin in the excited (down) state as a function of $t_w$, shown in **b** for a magnetic field of 4 T applied along different angles $\phi$ as indicated. Markers show measurements with statistical error bars, curves are fits to the formula $P_e(t_w) \propto (\exp(-Wt_w) - \exp(-\Gamma_{in}t_w))/(\Gamma_{in} - W)$, where the tunneling-in rate $\Gamma_{in}$ is determined independently (Supplementary Note 3). $W$ is thus extracted as the only fit parameter. Error bars are standard deviations from fitting to counts (Supplementary Note 3)

shown in Fig. 4b, the anisotropy is seen to decrease from about 16 at high fields towards one at fields below 1 T. This behavior displays spin relaxation with equal speed in both principal directions, thus indicating isotropic relaxation at low fields. Together with the $W \propto B^3$ scaling, these observations constitute unequivocal demonstration of HF-mediated spin relaxation.

In conclusion, we have demonstrated a spin relaxation time of up to $57 \pm 15$ s limited by HF-phonon spin relaxation in a single electron lateral GaAs quantum dot, exhibiting a $\propto B^3$ field scaling together with isotropic relaxation at fields below 1 T. At larger fields, the spin relaxation becomes strongly anisotropic, with $W_X/W_Y \sim 16$, and the B-field scaling follows a $W \propto B^5$ law. Using excited state spectroscopy, we determine the dot orbital energies, can extract the Rashba and linear Dresselhaus parameters from the B-field anisotropy of $W$, and simulate the HF induced spin relaxation $W$, in very good agreement with the experiment. While ramping the magnetic field from 0.6 T to about 10 T, the spin relaxation rate changes by a striking six orders of magnitude. Yet this is captured by the theory throughout the entire range—putting the model using a single set of parameters to a very stringent test. With the SOI parameters at hand, one can maximize the electric dipole spin resonance Rabi frequencies[7,33] in future experiments by optimizing geometry, with potentially large gains in qubit quality[34].

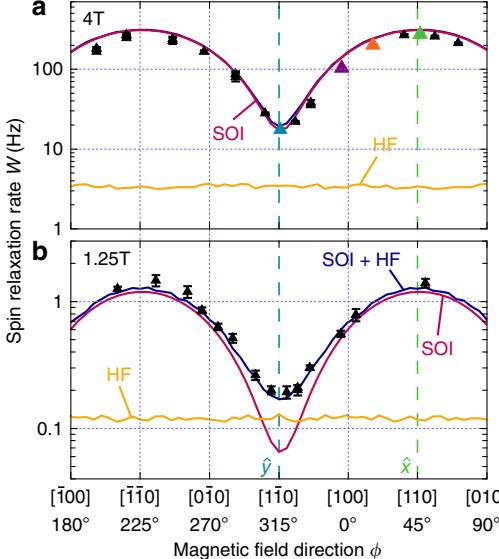

**Fig. 3** Spin relaxation anisotropy. Spin relaxation rate $W$ (triangles with error bars) for in-plane magnetic fields of **a** 4 T and **b** 1.25 T, as a function the field direction. The solid curves show the results from numerics taken into account only the SOI (red), only the HF interaction (orange), and both (dark blue). The ripples in curves from numerics are fluctuations due to finite statistics over random nuclear spin configurations. Error bars are standard deviations from fits to data as introduced in Fig. 2b

## Methods

**Sample and measurement**. The measurement was performed on a surface gate defined single-electron quantum dot formed in a GaAs 2D electron gas. The device was fabricated on a GaAs crystal, grown along the [001] crystal direction, with a GaAs/AlGaAs single heterojunction located 110 nm below the surface with density $2.6 \times 10^{11}$ cm$^{-2}$ and mobility $4 \times 10^5$ cm$^2$ V$^{-1}$ s$^{-1}$. The layout of the surface gates (Fig. 1a) is modified from that in ref. [18], and allows effective control of the dot shape. Negative gate voltages were applied on the gates to locally deplete the 2D gas and form a quantum dot in the center of the device (blue ellipse in Fig. 1a) and the adjacent charge sensor quantum dot (black dashed circle). The main dot is tuned to the single electron regime and tunnel coupled only to its left lead.

The single-electron quantum dot is capacitively coupled to the charge sensor, the conductance through which changes by 50–100% when adding or removing an electron to the main dot. Real-time detection of the dot charge state was realized by monitoring sensor dot current with a measurement bandwidth of 30 kHz obtained with a specially designed current preamplifier (Low-noise high-stability current preamp IF3602, Basel Electronics Lab) capable of handling capacitive input loads as appearing from the microwave filtering. The charge sensor bandwidth is limited by the low-pass filter of the preamp. For data acquisition as well as gate pulses, a National Instruments USB-6366 DAQ is used. The rectangular pulses are resistively coupled to a DC voltage offset with carefully matched impedance. Our lines show a resistance of about 40 Ω with a capacitance of about 5 nF dominated by the microwave filters[17], which leads to a technical bandwidth of about 1 MHz. To reduce the input capacitance induced noise on the IV-converter, microwave filters with a lower capacitance of 2 nF were used on those lines.

The main dot is electrically extremely stable due to excellent 2D gas material quality and allows control of the dot energy levels using a level positioning algorithm (Supplementary Note 2) for an extended period of time, which is crucial for long spin relaxation measurements. This feedback technique was regularly carried out throughout the measurements to compensate drift of the dot energy level with respect to chemical potential of the lead. Additionally, a feedback to compensate the drift of the sensor dot conductance was also performed regularly. Electron exchange processes with the reservoir[35] occurring during the charge and relax pulse step for long waiting times $t_w$ are detected by continuously monitoring the dot charge state and are removed from the data sets. This becomes an important factor particularly at low fields.

Lots of efforts have gone into operating at low electron temperatures[17,35–47], see ref. [38] for a recent review. The base temperature of the dilution refrigerator is $T_{base} \approx 25$ mK and the electron temperature is $T_{el} \approx 60$ mK, measured by probing the Fermi-Dirac distribution of the coupled lead. By heating to 300 mK where $T_{el} \approx T_{base}$, the Fermi-Dirac distribution method was also used to quantify the gate lever-arm. The sample was rotated (Attocube ANRv51/RES/LT/HV piezoelectric rotator) in a magnetic field up to 14 T applied in the plane of the 2D gas[48]. The out-of-plane magnetic field is determined by standard Hall effect measurements using van der Pauw configurations (Supplementary Note 1). The maximal misalignment of

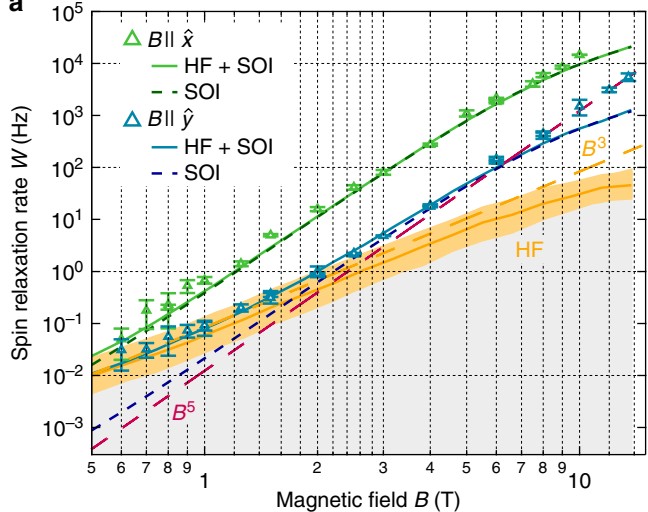

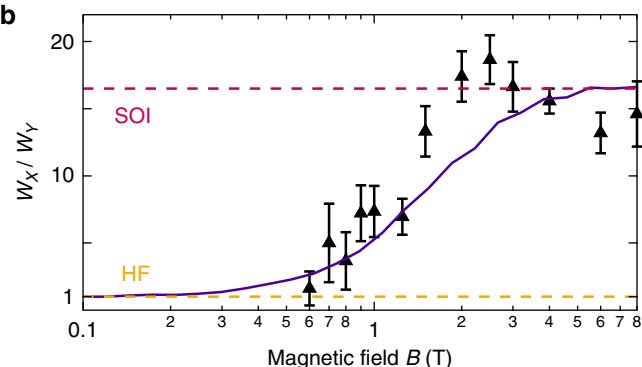

**Fig. 4** Hyperfine induced spin relaxation. **a** Spin relaxation rate $W$ for an in-plane magnetic field along the $\hat{x}$ direction (green, along [110]) and the $\hat{y}$ direction (blue, along [1$\bar{1}$0]) as a function of the field magnitude. The data are shown as triangles with error bars. Numerics considering various terms are shown as labeled. The pure $B^5$ scaling (red dash) and $B^3$ scaling (orange dash) are also given as a guide to the eye. The orange band around the HF curve indicates the statistical uncertainty due to a finite number of nuclear spin configurations used in the simulation. **b** The relaxation anisotropy $W_X/W_Y$ as a function of field magnitude. Experiment is shown as triangles with error bars, numerics with both SOI and HF as a solid curve, showing the transition to isotropic relaxation at low fields. Red dashed line is SOI theory only, orange dash at $W_X/W_Y = 1$ is the isotropic HF theory. A possible dip below the theory above $\gtrsim 6$ T could be due to the only remaining discrepancy between theory and experiment, occurring at hight fields (see main text). Error bars are fit errors

the in-plane magnetic field is 1.3°, thus the effect of the out-of-plane component is negligible[49].

With all these precautions, we achieve spin-state read-out fidelity of ~81% at low fields, and as high as 99% at higher fields. See Supplementary Note 4 for more details.

**The numerical model**. A microscopic model is used to describe the dot orbital spectroscopy and spin relaxation data. The implementation is based on an exact diagonalization of the electronic Hamiltonian which includes the kinetic energy with an anisotropic mass, a bi-quadratic (harmonic) confinement potential in the 2D plane, the Zeeman term, the linear and cubic spin–orbit terms, and the Fermi contact HF interaction with nuclear spins. This Hamiltonian is discretized in real space, typically on a grid of 100 by 100 points, with Dirichlet boundary conditions for the wavefunction. The resulting hermitian Hamiltonian matrix is diagonalized by the Arnoldi method using the ARPACK library, to obtain a few lowest eigenstates and the corresponding energies[50]. As an example, Fig. 1c, d (solid curves) shows the excitation energies calculated from such an exact spectrum as a function of the field. The spin relaxation rates are calculated by Fermi's golden rule using the exact spectrum, and bulk phonons coupled to electrons by deformation and

piezoelectric potentials. The rates denoted as "SOI" in the figures were obtained in the same way, but with the HF interaction excluded from the Hamiltonian. Similarly, the tag "HF" means that the spin–orbit terms were excluded.

The results from such a numerical procedure are expected to have a very high precision[51,52], in the sense of convergence (numerical stability), and also compared to analytical results in cases where the latter are known. As an example, the energies of the Fock-Darwin spectrum for our parameters match the analytical formulas up to errors well below 1 μeV. The errors stemming from the numerical procedures themselves are therefore expected to be completely negligible compared to errors induced by uncertainties of the used parameters, the true confinement shape, or the departures from the assumed simple forms of the spin–orbit, electron–phonon, and HF interactions. Whenever the Hamiltonian includes the HF interaction, the given relaxation rate is a geometric average of rates for 1000 configurations of static nuclear spins with random orientations (the approximation of unpolarized nuclei at infinite temperature). More details on the Hamiltonian and the numerical methods used to solve it are given in Supplementary Notes 5–9.

**Analytical results**. The following formulas reflect the main features of the relaxation rate important in our experiments. The relaxation rate due to transverse piezoelectric phonons and nuclear spins is

$$\Gamma_{\mathrm{HF}} \approx \frac{8(eh_{14})^2 I(I+1)A^2}{315\pi\hbar^2 m\rho c_t^5 N}\left(\frac{1}{E_x^3}+\frac{1}{E_y^3}\right)\left(g\mu_{\mathrm{B}}B\right)^3. \qquad (1)$$

It is isotropic and proportional to $B^3$. Replacing HF with spin–orbit effects leads to

$$\Gamma_{\mathrm{SOI}} \approx \frac{(eh_{14})^2}{210\pi m^2\rho c_t^5 l_{\mathrm{so}}^2}\left(\frac{1}{E_x^4}+\frac{1}{E_y^4}\right)\left(g\mu_{\mathrm{B}}B\right)^5 \times \left[\cos^2\xi(f_1+\epsilon f_2)+\sin^2\xi(f_3+\epsilon f_4)\right].$$
$$(2)$$

The rate grows as $B^5$ and is anisotropic, with the angular dependence described by

$$f_1 = 1 + \sin 2\vartheta \sin 2\phi,$$
$$f_2 = \sin 2\delta \sin 2\vartheta + \sin 2\delta \sin 2\phi + \cos 2\delta \cos 2\vartheta \cos 2\phi,$$
$$f_3 = 2,$$
$$f_4 = 2\sin 2\delta \sin 2\vartheta,$$
$$\epsilon = \left(E_x^{-4} - E_y^{-4}\right)/\left(E_x^{-4} + E_y^{-4}\right).$$

These formulas are derived in Supplementary Notes 5 and 6, where their generalized forms, including the effects of finite temperature, longitudinal phonons, and deformation electron–phonon potential, are also given.

The parameters in these equations are (values given for GaAs): piezoelectric potential $h_{14} = 1.4 \times 10^9$ V m$^{-1}$, nuclear spin $I = 3/2$, Fermi-contact interaction constant $A = 45$ μeV, effective mass $m = 0.067m_e$ with $m_e$ the electron mass in vacuum, material density $\rho = 5300$ kg m$^{-3}$, transverse acoustic phonon velocity $c_t = 3350$ m s$^{-1}$, Bohr magneton $\mu_{\mathrm{B}} = e\hbar/2m_e$. The number of nuclei in the dot $N \approx 8.3 \times 10^5$, the excitation energies $E_x = 2.33$ meV, $E_y = 2.61$ meV, the g-factor $g = -0.36$, and the angle of the dot potential axis with the [100] direction $\delta \approx 50.6°$, were fitted from spectral data such as in Fig. 1. The spin–orbit parameters $l_{\mathrm{so}} = 2.1$ μm and $\vartheta = 31°$, defined by writing the Rashba and Dresselhaus interaction strengths (see Supplementary Eq. (15) in Supplementary Note 5) as $\alpha = (\hbar/2ml_{\mathrm{so}})\cos\vartheta$, and $\beta = (\hbar/2ml_{\mathrm{so}})\sin\vartheta$, were fitted from the $T_1$ data shown in Figs. 3, 4. Finally, the magnetic field orientation is parameterized by writing **B** = $B[\cos\xi\cos\phi, \cos\xi\sin\phi, \sin\xi]$, referring to crystallographic coordinates.

**Code availability**. Computer codes and algorithms are available from the corresponding author upon reasonable request.

**Data availability**. The data that support the findings of this study are available in a Zenodo repository (https://doi.org/10.5281/zenodo.1241104 [not published yet])[53].

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

## Acknowledgements

We would like to thank V. Golovach and P. Scarlino for valuable inputs and stimulating discussions and M. Steinacher and S. Martin for technical support. This work was supported by the Swiss Nanoscience Institute (SNI), NCCR QSIT, Swiss NSF, ERC starting grant (DMZ), the European Microkelvin Platform (EMP). P.S. acknowledges support from CREST JST (JPMJCR1675), and JSPS Kakenhi Grant No. 16K05411.

## Author contributions

L.C.C., L.Y., P.S. and D.M.Z. designed the experiments, analysed the data and wrote the paper. L.C.C. and L.Y. processed the samples and performed the experiments. J.D.Z. and A.C.G. carried out the molecular beam epitaxy growth of the heterostructure. P.S. and D.L. developed and carried out the theoretical work and numerical modeling. All authors discussed the results and commented on the manuscript.

## Additional information

**Competing interests:** The authors declare no competing interests.

