## [Peer Review File · Nature Communications]

PEER REVIEW FILE

Reviewers' comments:

Reviewer #1 (Remarks to the Author):

The authors have addressed properly most of the questions posed and have improved the text accordingly.

I think however that it would be desirable a further investigation of the discrepancy between experiments and theory in Fig, 4b, concerning the observed dip in the relaxation anisotropy at B close to 6 T. An explanation in terms of possible interference between HF and SO at such high fields is not convincing in my opinion, an additional discussion would be desirable.

In summary, this work is interesting and relevant and it has been improved in the present version. A further discussion about the discrepancy mentioned above would strength the interpretation of the experimental results. Once it is included, my only concern, as I mentioned in my previous report is if it would not be more appropriate for a specialized journal.

Reviewer #2 (Remarks to the Author):

Authors have experimentally studied the hyperfine (HF) interaction mediated spin relaxation mechanism and the resultant long spin relaxation time of about 50 s in a gate-defined GaAs single quantum dot. Though the HF-induced mechanism has been predicted theoretically no experimental observation has been reported due to the technical difficulties. By overcoming the technical difficulties, lowering electron temperature, stable quantum dot gate operation, and in-situ sample rotation, the HF-induced spin relaxation has been verified. The quality of the experiment is very high and the theoretical analysis clearly supports the conclusion. The paper is well organized and is clearly written to reach to their conclusion. Of course the long spin relaxation time would give an impact for the general readers of Nature communications. Therefore, I would recommend the manuscript for publication in Nature communications.

I have two minor comments.

- 1) In the 4th paragraph in P.4, it would be better to change from “Fig 4 for $B > 9 T$.” to “Fig. 4a for $B > 9T$ ”.
- 2) The meaning of the orange belt for HF in Fig. 4a may not be explained.

Reviewer #3 (Remarks to the Author):

The authors addressed rigorously all the points raised in the different referee reports. In the present form, I recommend publication of the manuscript in Nature Com.

Responses to Referees

Font key

Referees: *italic*

Our reply: blue

Response to Referee 1

The authors have addressed properly most of the questions posed and have improved the text accordingly.

We thank the referee for this positive assessment.

I think however that it would be desirable a further investigation of the discrepancy between experiments and theory in Fig. 4b, concerning the observed dip in the relaxation anisotropy at B close to 6 T. An explanation in terms of possible interference between HF and SO at such high fields is not convincing in my opinion, an additional discussion would be desirable.

We thank the referee for pointing this out. This apparent discrepancy of the data and the theoretical model at high fields around 6 T appears not clearly statistically significant given the experimental error bars. One data point (6 T) is dipping somewhat below the theory (by one error bar), while the other point (8 T) in Fig. 4b is in agreement with the theory, within error bars. Nevertheless, we will consider the possibility of such a discrepancy at close to 6 T and above and provide a discussion of this possibility.

Assuming that such a dip of the experiment below the theory at and above 6T is present in Fig. 4b, we note that this is closely related to the only remaining discrepancy between theory and experiment at high fields $> \sim 6$ T which was pointed out in the discussion of Fig 4a. The relaxation rate for the slower direction rises somewhat above the theory, and thus the ratio of faster over slower rate would dip below the theory expectation. Thus the possible dip in Fig. 4b is related to this remaining high field discrepancy. This was discussed already in the revised manuscript of the previous submission, blue paragraph (9 lines long), left column, page 4. An understanding of this deviation needs further investigation which we cannot provide here within reasonable effort. However, the issue is irrelevant for the nuclear-induced relaxation which is taking place at much lower fields (below 1 to 2 T) and much longer times.

“A possible dip below the theory above ~ 6 T could be due to the only remaining discrepancy between theory and experiment, occurring at high fields (see main text)”

In summary, this work is interesting and relevant and it has been improved in the present version. A further discussion about the discrepancy mentioned above would strength the interpretation of the experimental results. Once it is included, my only concern, as I mentioned in my previous report is if it would not be more appropriate for a specialized journal.

We think that our manuscript is of interest for the broad audience of Nature Communications because GaAs based spin qubit devices are still among the most studied in the community and spin relaxation is currently limiting the read-out fidelity of these devices. We show that the spin relaxation rate is limited by the hyperfine interaction in the low field limit where most spin qubits are operated. This newly found

process leads to relaxation orders of magnitude faster than what was expected due to spin orbit interaction. Also, we show a spin relaxation time of almost *one minute*, the longest measured to date in any nanodevice.

Response to Referee 2

Authors have experimentally studied the hyperfine (HF) interaction mediated spin relaxation mechanism and the resultant long spin relaxation time of about 50 s in a gate-defined GaAs single quantum dot. Though the HF-induced mechanism has been predicted theoretically no experimental observation has been reported due to the technical difficulties. By overcoming the technical difficulties, lowering electron temperature, stable quantum dot gate operation, and in-situ sample rotation, the HF-induced spin relaxation has been verified. The quality of the experiment is very high and the theoretical analysis clearly supports the conclusion. The paper is well organized and is clearly written to reach to their conclusion. Of course the long spin relaxation time would give an impact for the general readers of Nature communications.

Therefore, I would recommend the manuscript for publication in Nature communications.

We thank the referee for this very positive assessment of our manuscript.

I have two minor comments.

1) In the 4th paragraph in P.4, it would be better to change from “Fig 4 for $B > 9 T$.” to “Fig. 4a for $B > 9 T$ ”.

We thank the referee for pointing this out. We also discussed about the discrepancy and decided to change the sentence to “(see the blue data points and theory curve in Fig. 4a for $B > 8 T$)”.

2) The meaning of the orange belt for HF in Fig. 4a may not be explained.

Indeed, we did not explain the orange band of the hyperfine model and thank the referee to bring this to our attention. We added the following sentence to the caption of Fig 4a:

“The orange band around the HF curve indicates the statistical uncertainty due to a finite number of nuclear spin configurations used in the simulation.”

Response to Referee 3

The authors addressed rigorously all the points raised in the different referee reports. In the present form, I recommend publication of the manuscript in Nature Com.

We thank the reviewer for this very positive evaluation.

Reviewers' Comments:

Reviewer #1 (Remarks to the Author):

The authors answers to my concerns are satisfactory. Therefore I consider the manuscript suitable for publication in Nature Communications.